# A revised marine fossil record of the Mediterranean before and after the Messinian Salinity Crisis

Konstantina Agiadi[1]*, Niklas Hohmann[2,3], Elsa Gliozzi[4], Danae Thivaiou[5,6], Francesca R. Bosellini[7], Marco Taviani[8,9], Giovanni Bianucci[10], Alberto Collareta[10], Laurent Londeix[11], Costanza Faranda[4], Francesca Bulian[12,13], Efterpi Koskeridou[6], Francesca Lozar[14], Alan Maria Mancini[14, 15], Stefano Dominici[16], Pierre Moissette[6], Ildefonso Bajo Campos[17], Enrico Borghi[18], George Iliopoulos[19], Assimina Antonarakou[6], George Kontakiotis[6], Evangelia Besiou[6], Stergios D. Zarkogiannis[20], Mathias Harzhauser[21], Francisco Javier Sierro[12], Angelo Camerlenghi[22], Daniel García-Castellanos[23]

[1] Department of Geology, University of Vienna, Josef-Holaubek-Platz 2, 1090 Vienna, Austria
[2] Utrecht University, Faculty of Geosciences, Department of Earth Sciences, Vening Meineszgebouw A, Princetonlaan 8a, 3584 CB Utrecht, Netherlands
[3] Institute of Evolutionary Biology, University of Warsaw, Krakowskie Przedmieście 26/28, 00-927, Warsaw, Poland
[4] Dipartimento di Scienze, Università Roma Tre, L.go S. Leonardo Murialdo, 1 - 00146 Roma, Italy
[5] Natural History Museum of Basel, Augustinergasse 2, 4001, Basel, Switzerland
[6] Department of Historical Geology and Palaeontology, Faculty of Geology and Geoenvironment, National and Kapodistrian University of Athens, Panepistimioupolis Zografou 15784, Athens, Greece
[7] Dipartimento di Scienze Chimiche e Geologiche, Università degli Studi di Modena e Reggio Emilia, Italy
[8] Institute of Marine Science - National Research Council, ISMAR-CNR, Via Gobetti 101, 40129 Bologna, Italy
[9] Stazione Zoologica 'Anton Dohrn', Villa Comunale, Via  Caracciolo, 80122, Napoli, Italy
[10] Dipartimento di Scienze della Terra, Università di Pisa, Pisa, Italy
[11] Université de Bordeaux /UMR 'EPOC' CNRS 5805, allée Geoffroy St-Hilaire, 33615 Pessac Cedex, France
[12] Department of Geology, University of Salamanca, Plaza de Los Caidos s/n, 37008, Salamanca, Spain
[13] Groningen Institute of Archaeology, University of Groningen, Postsraat 6, 9712, Groningen, the Netherlands
[14] University of Torino, Department of Earth Sciences, Via Valperga Caluso 35, 10125 Torino, Italy
[15] Department of Life and Environmental Science, Università Politecnica delle Marche, 60122 Ancona, Italy
[16] Museo di Storia Naturale, Università degli Studi di Firenze, Italy
[17] Sección de Paleontología, Museo de Alcalá de Guadaíra, Seville, Spain
[18] Società Reggiana di Scienza Naturali, Reggio Emilia, Italy
[19] Department of Geology, University of Patras, University Campus, 26504 Rio, Achaia, Greece
[20] Department of Earth Sciences, University of Oxford, Oxford, UK
[21] Natural History Museum, Burgring 7, 1010, Vienna, Austria
[22] OGS Istituto Nazionale di Oceanografia e di Geofisica Sperimentale, Trieste, Italy
[23] Geosciences Barcelona (GEO3BCN-CSIC), Solé i Sabarís s/n, Barcelona, Spain

*Correspondence to*: Konstantina Agiadi (konstantina.agiadi@univie.ac.at)

**Abstract.** The Messinian Salinity Crisis and its precursor events have been the greatest environmental perturbation of the Mediterranean Sea to date, offering an opportunity to study the response of marine ecosystems to extreme hydrological change and a large-scale biological invasion. The restriction of the marine connection between the Mediterranean and the Atlantic Ocean resulted in stratification of the water column and high-amplitude variations in seawater temperature and salinity already since the early Messinian. Here, we present a unified and revised marine fossil record of the Mediterranean (Agiadi et al., 2024) that covers the Tortonian stage, the pre-evaporitic Messinian and the Zanclean stage, and encompasses

23032 occurrences of calcareous nannoplankton, dinoflagellates, foraminifera, corals, ostracods, bryozoans, echinoids, mollusks, fishes, and marine mammals. This record adheres to the FAIR principles, it is updated in terms of taxonomy, and it follows the currently accepted stratigraphic framework. Based on this record, knowledge gaps are identified, which are due to spatiotemporal inconsistencies in sampling effort and the distribution of sedimentary facies, and the inherent differences in the preservation potential between the groups. Additionally, sampling bias in old records may have distorted the record in favor of larger, more impressive taxa within groups. This record is now ready to be used to answer both geological and biological questions about the Mediterranean Sea and beyond, and is amendable when new fossil data are brought to light.

## 1 Introduction

The Mediterranean Sea is ones of the areas of the world suffering today from an increase rate of warming as well as the immigration of warm-water non-indigenous species from the Red Sea through the Suez Canal (Galil, 2000; Marcott et al., 2013). The lack of time-series that are long enough to contain periods of time when climate was equally warm as future projections foretell or when the basin was connected to other oceanic basins prevents establishing thresholds and evaluating the resilience of the Mediterranean marine ecosystem to the anticipated change.

The Tortonian–Zanclean was a pivotal period in the evolution of the Mediterranean realm, predominantly due to the changes in the marine gateway configurations that affected the hydrological budget and oceanographic conditions in the basin, leading to the Messinian Salinity Crisis (MSC; Hsü et al., 1973). Even though the Neogene Mediterranean has been one of the most intensely researched geoscience topics in the last 150 years, its fossil record is largely fragmented and outdated, both in terms of taxonomy and stratigraphy. This has prevented assessing the implications of the MSC on the regional Mediterranean and global scale. Here, we provide a database of the revised marine fossil record of the Mediterranean basin, before and after the MSC. The fossil record during the MSC has been addressed in previous works (Carnevale et al., 2019; Carnevale and Schwarzhans, 2022) and is not included here because it is much more limited spatially and temporally and cannot be used to assess large-scale changes statistically. This open-access dataset adheres to the FAIR (findable, accessible, interoperable, and reusable) principles (Wilkinson et al., 2016). It can be enlarged and amended in the future, to enhance our understanding of the response of marine ecosystems to large-scale connectivity changes. It includes marine species occurrences in the Mediterranean Sea during the Tortonian age (13.8–7.25 Ma), the pre-MSC Messinian (7.25–5.97 Ma) and the Zanclean age (5.33–3.6 Ma) for the following groups: calcareous nannoplankton, dinocysts, foraminifera, corals, ostracods, bryozoans, echinoids, mollusks, fishes, and marine mammals.

In order to construct this dataset, we relied on the paleontological work conducted across the Mediterranean by generations of scientists, which included: mapping, field collection and taphonomic observations, systematic identification and documentation of the fossil material, chronostratigraphy and paleoenvironmental reconstructions. Most of these studies, which are available as separate papers dealing with each topic, were published over several years or even decades for each location. There have been a few reviews of the fossil record for some groups, but these are now in need of revision because

both the stratigraphic framework and the systematic schemes have changed since their publication. Such studies include Benson (1976a) for ostracods, Cita (1976) for planktic foraminifera, Sorbini & Tirapelle Rancan (1980) for Messinian fishes,

Moissette & Pouyet (1987) for bryozoans, Néraudeau et al. (2001) for irregular echinoids, and Monegatti & Raffi (2010) for Messinian gastropods and bivalves. None of these studies provided a dataset with the occurrences, but only reviewed the previous literature. This dataset is unique because it consists of georeferenced occurrences of species-level identified fossil material and includes level of uncertainty, which is necessary for quantifying ecological impacts at regional and global level.

## 2 Dataset structure

We collected the published fossil records of marine taxa in the Tortonian, the pre-evaporitic Messinian, and the Zanclean (Agiadi et al., 2024). For each record, the database includes the name of the taxon as in the original publication, the higher taxonomic group and the family it belongs to, the locality where it was found, the relative age of the sediments (separated into the three above categories), the publication(s) providing evidence for the occurrence, and the initials of the experts who collected and input the data (as given in the Authors Contribution statement) (Fig. 1).

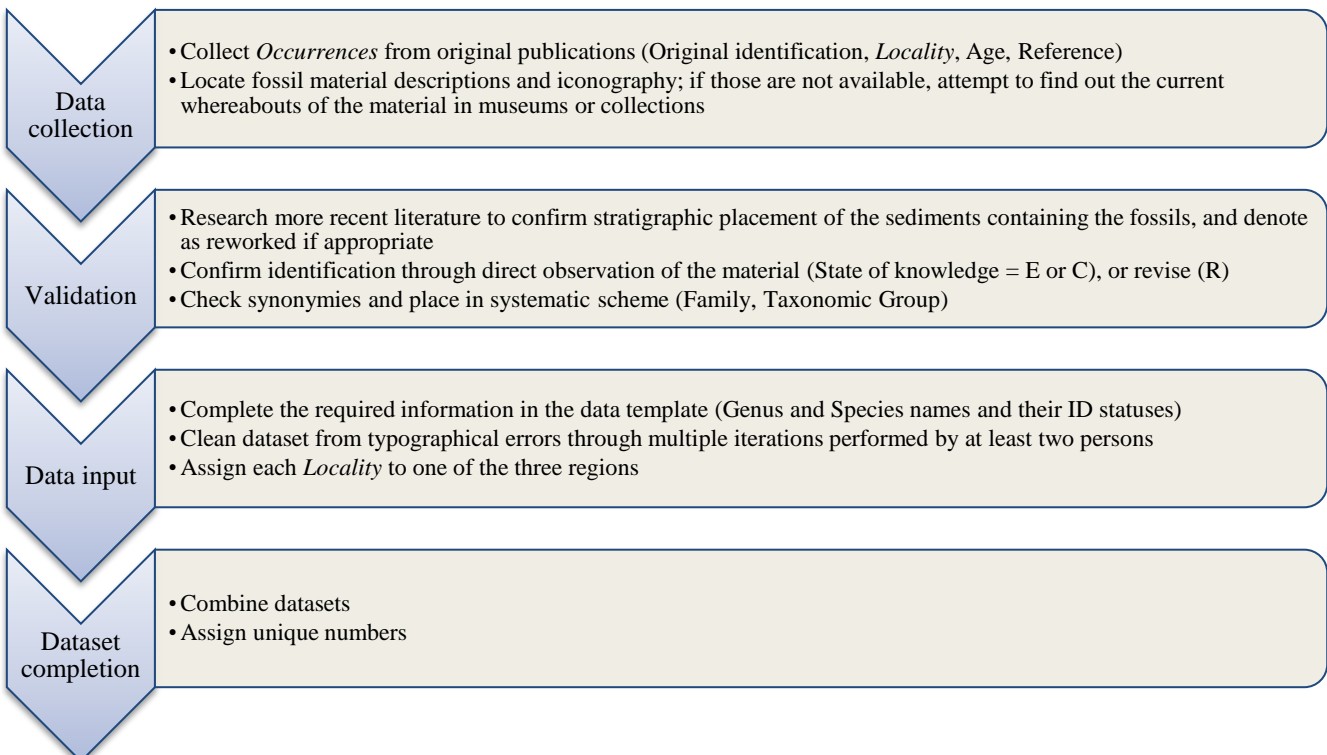

**Figure 1. Flowchart showing the steps followed to collect, validate, input and complete the dataset.**

An *Occurrence* was defined by the unique combination of three components: taxon, locality, age. To facilitate future analyses at the genus and species levels, since we reported fossil occurrences at all taxonomic levels, we indicated the genus and species identification status for each record. When the record was in open nomenclature, we indicated this as uncertain (U) identification status of the species and/or the genus. However, when a species was described as a different morphotype, but not named as a new species (e.g., sp. 1, sp. A etc.), we treated it as a separate species, if the corresponding expert(s) judged that the fossil material most probably belonged to a different, yet unnamed, species.

*Locality* refers to the distinct geographical site, where fossils were collected from. Often it was not possible to locate the origin of each record precisely, to the outcrop level, because several records were published many decades ago, and the outcrops had since then been destroyed and the authors are deceased. We therefore retained the most specific information we could find based on the description of the locality in the publication. The coordinates of the localities were obtained either from the occurrence publication or by estimation based on the locality description. Each locality was assigned to one of three regions based on their paleogeographic placement in the Western Mediterranean (wMed), the Eastern Mediterranean (eMed), and the Po Plain–Northern Adriatic (PoA), following the currently accepted paleogeographic data (Amadori et al., 2018; Steininger and Rögl, 1984). The PoA region developed as a paleoceanographic sub-basin of the Mediterranean in the Tortonian–Early Messinian, as evidenced by its distinct strontium isotopic signature (Cornacchia et al., 2021) and the absence of halite deposits. To facilitate comparisons, apart from the Messinian and Zanclean records, we placed also the Tortonian records from Piedmont and the Po Plain within the PoA region. Based on the Late Miocene paleogeographic evolution of Calabria and Sicily (Butler et al., 1995; Caracciolo et al., 2013; Henriquet et al., 2020), we included the records from these areas as part of the eMed, since the marine connection with the wMed was located near its present location, possibly along present southern Sicily (Micallef et al., 2019) or at the Sicily Channel (Malta Plateau; Bache et al., 2012). When the exact coordinates for a locality had not been published or when the locality was too vague (e.g. Libya), we included approximate coordinates based on the description of the locality in the original publication.

The validity of each record was then assessed by the corresponding expert(s), considering changes in the stratigraphic placement of the sedimentary formation from which it derived, new taxon synonymies or reassignment since the publication of the record, and previous misidentifications, leading to a revised name for that record, if necessary (Fig. 1). Taxonomy followed the World Register of Marine Species (WoRMS Editorial Board, 2024) and the systematic schemes of: Kroh and Smith (2010) for echinoids; Nelson et al. (2016) for fishes; Marx et al. (2016) for cetaceans; and Berta et al. (2018) for pinnipeds. The state of knowledge was accordingly noted in each case, as: E. if the identification was confirmed by the expert based on own sample examination, C. if the identification was confirmed by the expert based on photographs of the material that are available in the literature, R. if the identification was revised by the expert in the present study, and L. if the identification was not confirmed due to lack of access to the fossil material and absence of photographs in the literature, and the record relied only on the literature reference.

Regarding elasmobranchs, our dataset includes only sharks, because there is not enough information on the dental morphology of most extant rays (batomorphs). Many batomorph teeth forms are rarely figured in the literature. Teeth, often

isolated, are the commonest elasmobranch fossils, and comparative information from extant species is necessary to identify them. For several genera of rays, species-level identifications of fossil teeth are further hindered by the broad intra-generic variability. In addition, whilst some historical collections of sharks teeth from the Mediterranean Miocene and Pliocene have

125 been extensively reviewed, providing at least an updated nomenclatural framework to build upon, the same does not apply for batomorphs.

## 3 Fossil record overview

The assembled dataset comprised 23032 occurrences including 4903 species, 1755 genera and 640 families (Table 1; Fig. 2).

**Table 1. Overview of the collected fossil dataset in the three regions: Eastern Mediterranean (eMed), Western Mediterranean (wMed), Po Plain-Northern Adriatic (PoA). The group 'other molluscs' includes scaphopods, chitons and cephalopods.**

| Taxonomic group(s) | # of Occurrences | | | | Species | Genera | Families | # of Localities | | | |
|---|---|---|---|---|---|---|---|---|---|---|---|
| | all | eMed | wMed | PoA | | | | all | eMed | wMed | PoA |
| Marine mammals | 164 | 65 | 60 | 39 | 27 | 25 | 19 | 114 | 42 | 47 | 25 |
| Sharks | 216 | 55 | 110 | 51 | 50 | 47 | 26 | 37 | 7 | 23 | 7 |
| Bony fishes | 1601 | 365 | 695 | 541 | 426 | 265 | 127 | 70 | 22 | 32 | 16 |
| Gastropods | 5909 | 1113 | 2056 | 2740 | 1745 | 532 | 148 | 129 | 50 | 36 | 43 |
| Bivalves | 3671 | 701 | 2173 | 797 | 473 | 234 | 70 | 161 | 42 | 101 | 18 |
| Bryozoans | 833 | 319 | 514 | 0 | 263 | 169 | 83 | 6 | 3 | 3 | 0 |
| Echinoids | 217 | 27 | 173 | 17 | 109 | 38 | 22 | 42 | 12 | 22 | 8 |
| Corals | 902 | 235 | 514 | 153 | 133 | 57 | 23 | 242 | 73 | 149 | 20 |
| Ostracods | 4500 | 2215 | 938 | 1347 | 1084 | 168 | 29 | 124 | 59 | 37 | 29 |
| Other molluscs | 191 | 26 | 53 | 112 | 77 | 30 | 16 | 23 | 10 | 2 | 11 |
| Benthic foraminifera | 560 | 202 | 233 | 125 | 116 | 63 | 43 | 20 | 6 | 10 | 4 |
| Planktic foraminifera | 1306 | 651 | 468 | 187 | 97 | 17 | 3 | 79 | 37 | 29 | 13 |
| Dinoflagellate cysts | 1011 | 599 | 298 | 114 | 171 | 70 | 13 | 37 | 20 | 16 | 2 |
| Calcareous nannoplankton | 1951 | 971 | 707 | 273 | 132 | 40 | 18 | 56 | 26 | 20 | 10 |
| Total | 23032 | 7544 | 8992 | 6496 | 4903 | 1755 | 640 | 835 | 279 | 423 | 135 |

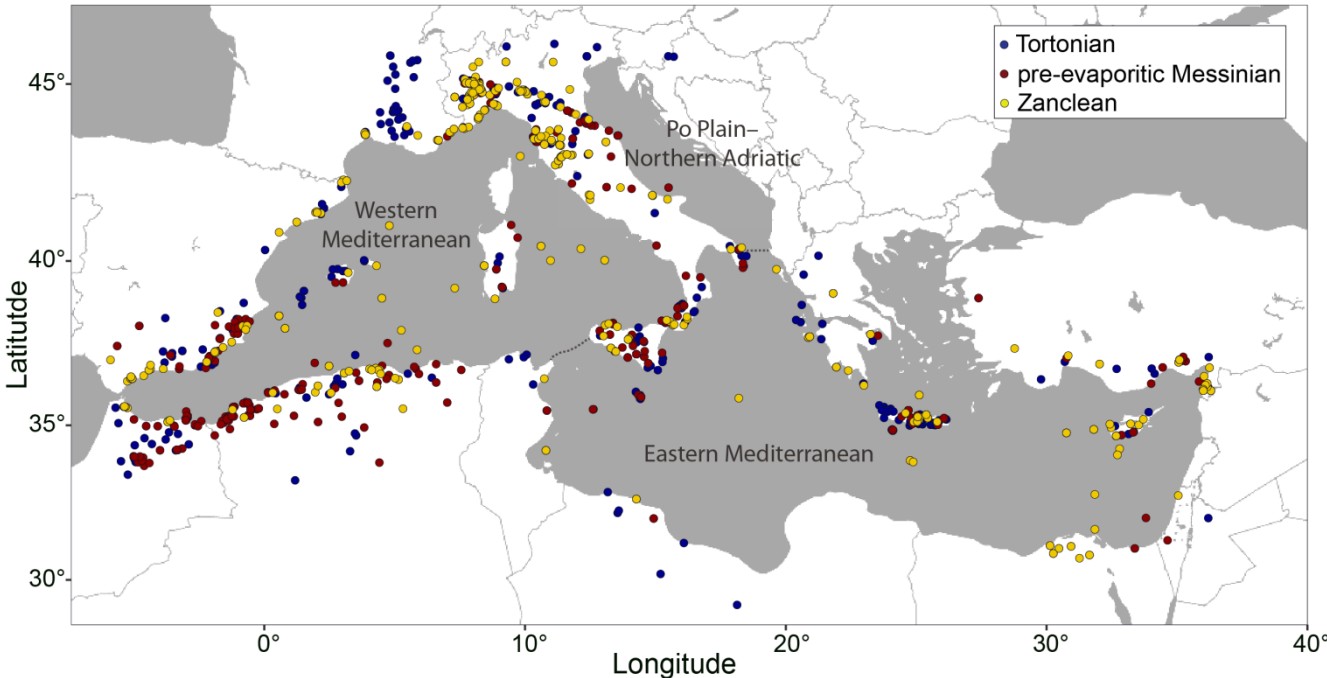

**Figure 2. Mediterranean map with the localities included in this record. Dashed lines show the borders between the three regions: Western Mediterranean, Eastern Mediterranean, Po-Plain–Northern Adriatic. The coordinates of the localities were obtained from the original publications, whenever possible. The map was produced using ggmap (Kahle and Wickham, 2013).**

## 4 Record gaps

First of all, some taxonomic issues remain that can only be addressed with further basic research and accumulation of new fossil material through fieldwork. Several species identification could not be confirmed (State of knowledge = L) because the material was never figured. Even new species were established, on occasion, in earlier works without descriptions or illustrations (*nomina nuda*; Doruk, 1979).

Some gaps are detected in the fossil record. Spatiotemporal gaps are attributed to: a) the spatial distribution of outcrops of Late Miocene–Early Pliocene deposits (Mascle and Mascle, 2012); and b) socioeconomic and political conditions favoring research in the northwestern Mediterranean countries (Cappelletto and et al., 2021). The spatial distribution of the localities within the Mediterranean is strongly skewed toward the West and the North (Figs. 2–4; e.g., for molluscs Monegatti and Raffi, 2010). However, there are notable exceptions. For example, most Tortonian data on gastropods derive from the eastern Mediterranean, whereas almost all Messinian and Zanclean localities are in the western sub-basin or the Po Plain-Northern Adriatic region (Table 1). On the other hand, fossils are rarer the larger the animal is. Only a single diverse shark tooth assemblage is currently known from the pre-evaporitic Messinian of the Mediterranean Basin, from the vicinities of Oran

(Algeria; Arambourg, 1927). Similarly, the fossil record of pinnipeds from the Late Miocene–Early Pliocene of the Mediterranean Basin only consists of two Monachinae (Phocidae) species: *Messiphoca mauritanica* from the Messinian (during the MSC) of Algeria (Muizon, 1981) and fragmentary remains referred of *Pliophoca* cf. *etrusca* from southern France, Italy and Spain (Berta et al., 2015).

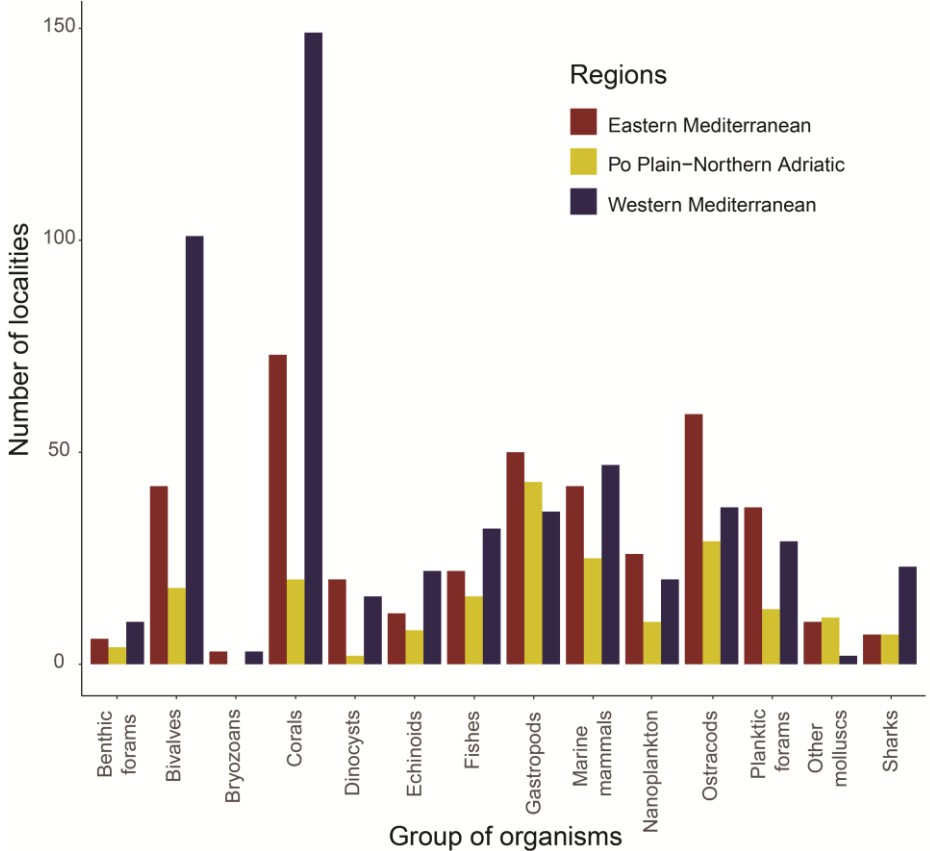

**Figure 3. Number of localities from where data has been included in the database, by group of organisms and region.**

Moreover, large species, especially those with more robust remains, have generally been favored in the fossil record of some groups, such as molluscs, as they were generally easier to sample (Dominici et al., 2020). This was in spite of the fact that a large proportion of the species in modern biodiversity hotspots are smaller than 4 mm (Bouchet et al., 2002). Particularly for
the case of molluscs, bivalves, contrary to gastropods, include calcitic forms that have a higher preservation potential than aragonitic forms. In contrast to gastropods, there are many more pre-evaporitic Messinian localities with rich bivalve faunas than either Tortonian and Zanclean localities. This is largely due to a few large expeditions in North Africa (Cornée et al., 2014; el Kadiri et al., 2010; Merzeraud et al., 2019), which yielded however only faunal lists of large-sized, mostly calcitic forms extracted from calcarenites, a lithology underrepresented in Zanclean collections (Dominici and Forli, 2021; Dominici
et al., 2019).

Important gaps in the Late Miocene–Early Pliocene marine fossil record of the Mediterranean derive from the uneven distribution of the different facies. This affects especially the records of benthic organisms, whose distribution depends on the type of substratum. In the case of molluscs, knowledge of the onshore-offshore facies gradient is incomplete (Dominici and Forli, 2021; Dominici et al., 2019). As a result, for gastropods, Tortonian and Zanclean data represent a wider facies range, from onshore to offshore siliciclastic, whereas the pre-evaporitic Messinian data are mainly confined to open shelf mudstones. As a result, littoral and bathyal taxa of the pre-evaporitic Messinian may be underrepresented in the database. Moreover, the Messinian mollusc data include records from hybrid carbonates (e.g., Dominici et al., 2019), which are lacking in the Zanclean (Dominici and Forli, 2021). Such gaps do not appear consistently across groups, however: for ostracods, both shallow and deeper siliciclastic facies are represented from the three geographic areas and the three stratigraphic intervals, rendering the dataset rather more complete.

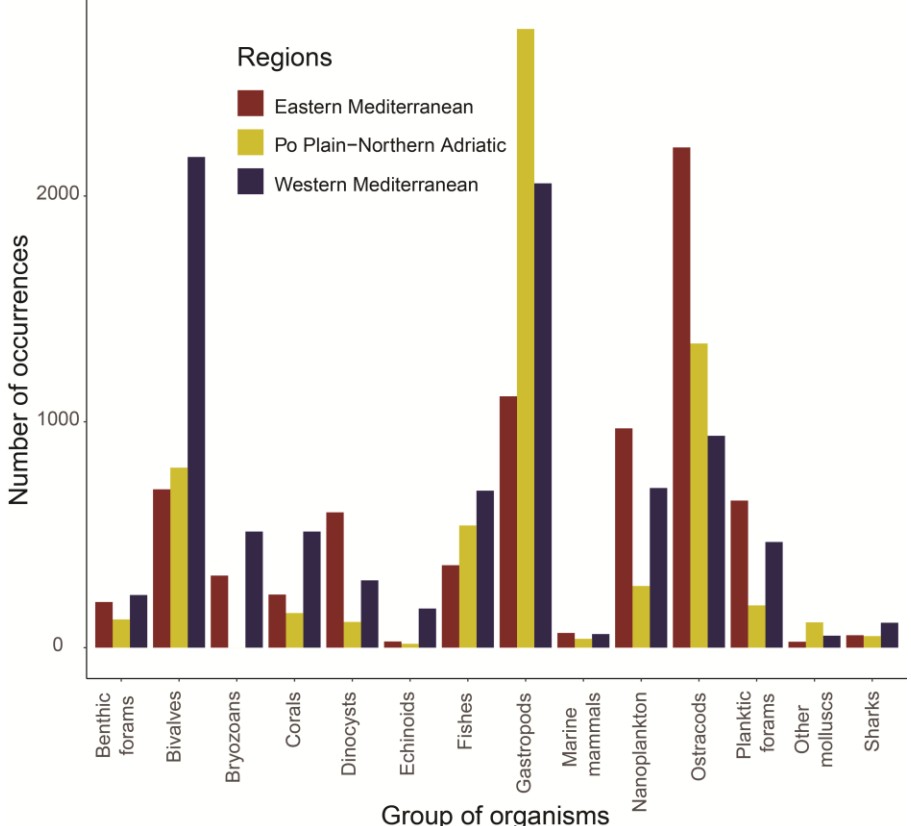

**Figure 4. Number of occurrences from where data has been included in the database, by group of organisms and region.**

Regarding stratigraphic gaps, the main issue when assembling the dataset derived from the fact that in some cases the chronostratigraphic framework provided in the initial publication too broad (e.g. Late Miocene or Pliocene in general) or questionable because it was based on lithostratigraphic correlation (e.g., Benson, 1976b; Doruk, 1979; Sissingh, 1972). In

the case of echinoids for example, many finding localities in the rich Italian Pliocene are generically reported in the geologic maps and publications as 'Zanclean–Piacenzian', thus preventing the precise temporal collocation of numerous citations from the Argille Azzurre Formation of the PoA region (e.g., Airaghi, 1901; Botto-Micca, 1896) and from Tuscany (e.g., Desor, 1858; Meneghini, 1862). There are two different problems here: old citations rarely indicated the precise stratigraphic position, whereas several precise layers yielding echinoids have not been studied yet by modern methods. For the Pliocene, records were only included in the database if they could be securely placed in the Zanclean based on the current knowledge on the stratigraphic placement of the deposits they were recovered from. The result is a limited record for the Messinian and possibly the Zanclean, both potentially leading to an underestimation of echinoid diversity.

**5 Data availability**

The dataset described in the paper is openly accessible under a CC BY license at https://doi.org/10.5281/zenodo.13358435 (Agiadi et al., 2024).

**6 Code availability**

No code was used for assembling the database. The code to generate Table 1 and Figs. 2–4 is available at https://doi.org/10.5281/zenodo.13358742 (Hohmann and Agiadi, 2024).

**7 Conclusions and Outlook**

This dataset includes a Tortonian–Zanclean marine fossil record of the Mediterranean before and after the Messinian Salinity Crisis. Initially, this dataset can be used to quantify the impact on marine biota of the Messinian Salinity Crisis, which was the greatest paleoenvironmental perturbation of the Mediterranean. The Late Miocene–Early Pliocene Mediterranean fossil record is invaluable, not only for large-scale paleobiogeographic studies, but also for evaluating the indigenous/non-indigenous status of tropical marine species detected today in the eastern Mediterranean, establishing resilience thresholds for marine organisms and their ecosystems, and investigating evolutionary dynamics, particularly of higher trophic-level groups. Nevertheless, we highlight the need for further targeted sampling expeditions and collaborative paleontological investigations facilitated by science diplomacy (Soler and Perez-Porro, 2021) to fill in these spatial gaps in the fossil record of the Mediterranean.

**Author contribution**

**Konstantina Agiadi**: Conceptualization, Data curation, Funding acquisition, Investigation (data collection: bony fishes), Methodology, Project administration, Writing – original draft preparation. **Niklas Hohmann**: Data curation, Formal

analysis, Methodology, Writing – review & editing. **Elsa Gliozzi**: Investigation (data collection: ostracods), Writing – review & editing. **Danae Thivaiou**: Data curation, Investigation (data collection: bivalves, gastropods), Writing – review & editing. **Francesca Bosellini**: Investigation (data collection: corals), Writing – review & editing. **Marco Taviani**: Investigation (data collection: corals), Writing – review & editing. **Giovanni Bianucci**: Investigation (data collection: marine mammals), Writing – review & editing. **Alberto Collareta**: Investigation (data collection: sharks), Writing – review & editing. **Laurent Londeix**: Investigation (data collection: dinoflagellates), Writing – review & editing. **Costanza Faranda**: Investigation (data collection: ostracods), Writing – review & editing. **Francesca Bulian**: Investigation (data collection: benthic foraminifera), Writing – review & editing. **Efterpi Koskeridou**: Investigation (data collection: bivalves, gastropods), Writing – review & editing. **Francesca Lozar**: Investigation (data collection: calcareous nannoplankton), Writing – review & editing. **Alan Maria Mancini**: Investigation (data collection: calcareous nannoplankton), Writing – review & editing. **Stefano Dominici**: Investigation (data collection: bivalves, gastropods), Writing – review & editing. **Pierre Moissette**: Investigation (data collection: bryozoans), Writing – review & editing. **Ildefonso Bajo Campos**: Investigation (data collection: echinoids), Writing – review & editing. **Enrico Borghi**: Investigation (data collection: echinoids), Writing – review & editing. **George Iliopoulos**: Investigation (data collection: marine mammals), Writing – review & editing. **Assimina Antonarakou**: Investigation (data collection: planktic foraminifera), Writing – review & editing. **George Kontakiotis**: Investigation (data collection: planktic foraminifera), Writing – review & editing. **Evangelia Besiou**: Investigation (data collection: planktic foraminifera), Writing – review & editing. **Stergios D. Zarkogiannis**: Investigation (data collection: planktic foraminifera), Writing – review & editing. **Mathias Harzhauser**: Investigation (data collection: bivalves, gastropods), Writing – review & editing. **Francisco Javier Sierro Sànchez**: Investigation (data collection: benthic foraminifera), Writing – review & editing. **Angelo Camerlenghi**: Conceptualization, Funding acquisition, Writing – review & editing. **Daniel García-Castellanos**: Conceptualization, Funding acquisition, Writing – review & editing.

**Competing interests**

The authors declare that they have no conflict of interest.

**Acknowledgements**

The authors would like to thank Dr. Fadl Raad and the Anonymous Reviewer for their constructive comments. Collaboration for this research was facilitated by the COST Action CA15103 "Uncovering the Mediterranean salt giant" (2016–2020). This research was funded in whole, or in part, by the Austrian Science Fund (FWF) [Grant DOI 10.55776/V986, available via https://www.fwf.ac.at/en/discover/research-radar] (PI: K. Agiadi). For the purpose of open access, the author has applied a CC BY public copyright license to any Author Accepted Manuscript version arising from this submission. This is Ismar-

CNR, Bologna, scientific contribution n. 2090. Konstantina Agiadi was also supported for this work by Greek national funds
and the European Social Fund through the action "Postdoctoral Research Fellowships" of the Greek National Scholarships
Foundation, project "Comparative study of the Messinian Salinity Crisis effect on the eastern Ionian and northern Aegean
ichthyofauna" (2017–2019). This research was co-funded by the European Union (ERC, MindTheGap, StG project no
101041077). Views and opinions expressed are exclusively those of the author(s) and do not necessarily reflect those of the
European Union or the European Research Council. Neither the European Union nor the granting authority can be held
responsible for them. Elsa Gliozzi and Costanza Faranda were funded by MIUR-Italy, Department of Excellence grant,
Article 1, Paragraph 337, law 232/2016. This study was partially funded by the European Commission through ITN
SaltGiant (Horizon2020-765256).

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
