# Peer review of "A revised marine fossil record of the Mediterranean before and after the Messinian Salinity Crisis"

_Earth System Science Data, 2024_

## Author Response (AR1)

**Response to Reviewer Comments:**

Reviewer #1 (Dr. Fadl Raad)

I am delighted to see this comprehensive review of the Mediterranean fossil record surrounding the MSC. The database is extensive and excellently accommodates the documented biodiversity records. This study makes a significant contribution by providing a revised and comprehensive fossil record for the Mediterranean before and after the MSC. The adherence to FAIR principles enhances the dataset's utility and accessibility for future research. Discussion could be slightly expanded to include broader implications and future research opportunities. Including additional plots, such as pie charts or histograms, could help visualize the distribution and diversity of the fossil records within the database, making the data more accessible and interpretable for readers. Agiadi et al., dataset manuscript presents a valuable dataset and a thorough analysis of the Mediterranean marine fossil record. With some very minor (mostly technical) revisions, it will be a very good contribution to the field of geosciences.

Thank you for your kind evaluation of our work. We address your comments below point-by-point, and we have incorporated the changes you suggested directly on the text as well.

Specific comments:

The introduction could be enhanced by briefly discussing the broader implications of understanding the MSC for current and future ecological and geological studies.

We added two sentences in the Introduction:

"This has prevented assessing the implications of the MSC on the regional Mediterranean and global scale."

and

"This dataset is unique because it consists of georeferenced occurrences of species-level identified fossil material and includes level of uncertainty, which is necessary for quantifying ecological impacts at regional and global level."

Consider providing a flowchart or diagram to visually represent the dataset structure and the process of data compilation and validation. More details on how the taxonomy was updated and standardized would be helpful for reproducibility.

Added. Thank you for the suggestion.

Authors could Include more specific information on the criteria used for reassigning or confirming taxonomic classifications, and discuss any potential biases introduced by the reliance on historical records and how these were mitigated.

In general, we followed WoRMS, and specific systematic schemes were adopted: Kroh & Smith (2010) for echinoids, Nelson et al. (2016) for fishes, Marx et al. (2016) for cetaceans, and Berta et al (2018) for pinnipeds.

The new Figure 1 shows the steps followed for the validation of the data. For those occurrences where there were neither descriptions nor figured specimens in the literature, and it was not possible to directly observe material in fossil collections, the records were kept with the indication *L* for the State of knowledge.

Indeed, this poses in some cases a long-standing issue, which can only be solved with further collection of fossil specimens. To highlight this, we added the following paragraph in the Discussion:

"First of all, some taxonomic issues remain that can only be addressed with further basic research and accumulation of new fossil material through fieldwork. Several species identification could not be confirmed (State of knowledge = L) because the material was never figured. Even new species were established, on occasion, in earlier works without descriptions or illustrations (*nomina nuda*; Doruk, 1979)."

And the following text to address this in the Dataset Structure section:

"Regarding elasmobranchs, our dataset includes only sharks, because there is not enough information on the dental morphology of most extant rays (batomorphs). Many batomorph teeth forms are rarely figured in the literature. Teeth, often isolated, are the commonest elasmobranch fossils, and comparative information from extant species is necessary to identify them. For several batomorph genera, species-level identifications of fossil teeth are further hindered by the broad intrageneric vaiability. In addition, whilst some historical collections of sharks teeth from the Mediterranean Miocene and Pliocene have been extensively reviewed, providing at least an updated nomenclatural framework to build upon, the same does not apply for batomorphs."

The results section could be improved by including more comparative analysis with previous studies to highlight the advancements made by this revised record.

To highlight the differences between this and previous studies on the impact of the MSC on marine organisms, we added the following paragraph in the Introduction:

"In order to construct this dataset, we relied on the paleontological work conducted across the Mediterranean by generations of scientists, which included: mapping, field collection and taphonomic observations, systematic identification and documentation of the fossil material, chronostratigraphy and paleoenvironmental reconstructions. Most of these studies, which are available as separate papers dealing with each topic, were published over several years or even decades for each location. There have been a few reviews of the fossil record for some groups, but these are now in need of revision because both the stratigraphic framework and the systematic schemes have changed since their publication. Such studies include Benson (1976a) for ostracods, Cita (1976) for planktic foraminifera, Sorbini & Tirapelle Rancan (1980) for Messinian fishes, Moissette & Pouyet (1987) for bryozoans, Néraudeau et al. (2001) for irregular echinoids, and Monegatti & Raffi (2010) for Messinian gastropods and

bivalves. None of these studies provided a dataset with the occurrences, but only reviewed the previous literature. This dataset is unique because it consists of georeferenced occurrences of species-level identified fossil material and includes level of uncertainty, which is necessary for quantifying ecological impacts at regional and global level."

Including additional plots, such as pie charts or histograms, could help visualize the distribution and diversity of the fossil records within the database, making the data more accessible and interpretable for readers.

We added Figures 3 and 4 to illustrate the distribution of the data and help visualize the fossil record.

Figures and table can be referenced more in the text.

Done

Maybe the authors could suggest specific future research directions or questions that could be addressed using this dataset.

Initially, this dataset has been used to quantify the impact on marine biota of the Messinian Salinity Crisis, which was the greatest paleoenvironmental perturbation of the Mediterranean (Agiadi et al., 2024a, b). The Late Miocene–Early Pliocene Mediterranean fossil record is invaluable, not only for large-scale paleobiogeographic studies, but also for evaluating the indigenous/non-indigenous status of tropical marine species detected today in the eastern Mediterranean, establishing resilience thresholds for marine organisms and their ecosystems, and investigating evolutionary dynamics, particularly of higher trophic-level groups.

Reviewer #2

I want to congratulate with the authors for the excellent job that they made. The data set about the marine fossil record of the Tortonian-Zanclean interval (excluding the Messinian salinity crisis record) will be for sure very useful for comparing the effects of the progressive restriction of the Atlantic-Mediterranean gateways and more in general of the Messinian salinity crisis on marine biota. The text is easy to read and well structured. I had a look at the excel files in the data section and they seem to be easy to access. The only suggestion that I have is to add some explicative figures (pie charts, histograms etc.) showing the main changes suffered by the considered fossil groups in this critical interval of the history of the Mediterranean Basin.

Thank you for this kind assessment of our work. We have added more figures and plots to visualize the dataset assembly procedure and to describe our dataset. A detailed statistical analysis of this dataset to reveal the impact of the Messinian salinity crisis on marine biota is published now elsewhere.

**List of changes:**

- Revised the text at the points indicated by Reviewer #1.
- Added the data now made available by Maria Triantaphyllou in the dataset, thus addressing her Community Comment, and revised Table 1 accordingly.
- Added a flowchart (Figure 1) showing the steps followed to assemble the dataset.
- Added Figures 3 and 4 to show better illustrate the dataset.

---

## Referee Report (RR1)

Dear editor, dear authors,

First of all, I would like to congratulate the authors on their excellent work. The manuscript is a valuable contribution to the field, and I appreciate the effort that has gone into it.

I would also like to express my thanks to the authors for thoroughly addressing all the reviews and comments I provided. I am satisfied with the revisions, and from my side, I believe the article is now ready for publication as it stands.

Thank you for your hard work and dedication in improving the manuscript. I look forward to seeing the final version published.

Best regards,

Fadl Raad